# Clinical Evolution of Acute Vestibular Syndrome: Longitudinal Retrospective Analysis of Epidemiological Data and Prognostic Factors for Recovery

**DOI:** 10.3390/jpm13030407

**Published:** 2023-02-24

**Authors:** Pasqualina M. Picciotti, Roberta Anzivino, Jacopo Galli, Francesco Franceschi, Guido Conti, Benedetta Simeoni, Marcello Covino

**Affiliations:** 1Department of Head and Neck, Otolaryngology Institute, Fondazione Policlinico Universitario A. Gemelli IRCCS Università Cattolica del Sacro Cuore, 00168 Rome, Italy; 2Otolaryngology Unit, Sensorineural Department, Di Venere Hospital, ASL BARI, 70131 Bari, Italy; 3Department of Emergency Medicine, Fondazione Policlinico Universitario A. Gemelli IRCCS Università Cattolica del Sacro Cuore, 00168 Rome, Italy

**Keywords:** dizziness, vertigo, acute vestibular syndrome, emergency department

## Abstract

We evaluated the clinical features of patients admitted to the ED with acute vertigo, verifying symptoms after one month and one year to establish epidemiological factors and predictors of resolution. We reviewed 233 records of patients admitted to ED for acute vertigo (125 F and 85 M, mean age 56.12 years). We analyzed the correlation between time of resolution (at one month and one year) and symptoms duration (subjective and/or objective vertigo, instability, cervical pain, audiological, neurological, and neurovegetative symptoms), comorbidities, and therapies, the result of the clinical and instrumental vestibular examination. Resolution of acute vertigo occurred in 81.1%, while persistence of vertigo one year after ED access was reported in 18.8%. There were 135 patients who recovered in one month. The presence of instability, auditory and neurovegetative symptoms, and neck pain represents a significant factor for recovery within one year. Age over 65 and a history of hypertension are associated with a worst recovery. Patients with spontaneous Nystagmus or positive HIT showed a significant difference in symptoms recovery within one month and one year. Presence of positional Nystagmus represents a positive prognostic factor. Our findings emphasize the importance of clinical evaluation of the acute vertigo, helping the clinicians to define central or peripheral diagnosis and predict the resolution of vertigo.

## 1. Introduction

Dizziness and vertigo are some of the most common reasons for seeking medical help. They require interdisciplinary cooperation, and they are common presenting symptoms in the Emergency Department (ED), accounting for 1–10% of all ED visits [1,2,3,4].

Duration of symptoms is variable, and it can range from a few seconds to a few days. All conditions constituted by acute dizziness (with or without audiological symptoms), associated with nausea and/or vomiting, enduring a day or more, are included in the term “acute vestibular syndrome” [5,6]. Potential causes range from benign to deadly, and they can be classified as otoneurological (vestibular), medical (cardiovascular and metabolic especially), and psychiatric disorders. However, peripheral vestibular system disorders represent the most important cause, as well as the best recognizable ones. 

On the other hand, it is worth noting that acute vertigo could be the manifestation of central nervous system pathologies such as cerebrovascular disease (ischemic stroke in the brainstem or cerebellum), mimicking benign peripheral causes [7,8].

Several papers have been published about the acute vestibular syndrome in order to investigate the pathophysiological process underlying vertigo and especially define exclusion criteria for stroke. Diagnosis in the Emergency Department is based on the systematic analysis of the clinical history (onset and main characteristics of vertigo or dizziness, clinical course of symptoms, modulating and critical factors, associated symptoms, comorbidities, and therapies), clinical examination of the vestibular, ocular, motor, and cerebellar systems as well as basic monitoring (vital signs, ECG, blood tests).

Many clinical and diagnostic tools have already been proposed in the literature in order to assess patients with vertigo in ED [6,9,10,11], but a complete clinical vestibular evaluation is really more reliable and competent to accurately identify the origin (central or peripheral) of symptoms. However, the presence or absence of Nystagmus in most patients presenting acute dizziness is not considered a reliable sign to guide conclusive diagnosis and consequent decision-making, thus providing use and sometimes overuse of neuroimaging tests.

Systematic reviews have highlighted the importance of accurately assessing the risk of dangerous disorders, particularly ischemic stroke, and have emphasized the utility of a focused history and physical examination in these patients [12,13,14]. Being that these data are often non-conclusive for diagnosis and consequent decision-making, another debated topic is the overuse of neuroimaging tests in clinical practice. Neuroimaging should be performed under these conditions: detection of central focal motor or neurological signs on clinical examination, aphasia, and significant major cardiovascular risk [12].

So far, many reports have formulated several hypotheses about topodiagnosis in vertigo. However, epidemiological features, long-term symptoms, and predictive factors of resolution in patients with acute vestibular syndrome have been overlooked so far. 

The aim of our observational study was to evaluate several clinical features (symptoms and signs, age and sex, presence of comorbidities, therapies, and radiological investigation) of patients admitted to the ED with acute vertigo and to verify the presence of symptoms after one month and one year, in order to investigate the epidemiological factors and predictors of resolution most frequently associated.

## 2. Materials and Methods

We reviewed 233 medical records of patients admitted to ED with symptoms of acute vertigo syndrome. The 125 subjects were males (59.5%), and 85 were females (40.5%) aged from 11 to 92 years (mean age: 56.12 years). 

All patients were clinically evaluated by Emergency Department doctors and Otolaryngologists with particular expertise in vertigo patient evaluation. 

In the clinical history, we included different information: subjective and/or objective vertigo, instability, audiological (hearing loss, fullness, tinnitus), neurological (paraesthesia, loss of vision or strength), or neurovegetative symptoms (nausea, vomiting), cervical pain and elevated blood pressure. We also evaluated symptoms duration (>1 day) and previous episodes of vertigo. Regarding comorbidities, we considered recent airway infection (<1 month), history of hypertension, cardiovascular disease (excluding hypertension), diabetes, and central nervous system diseases. Therapies prior to ED access taken into account were Anti-histaminic, Neuroleptic, Steroids, and ASA. Patients were subjected to clinical vestibular assessment in our vestibular service by means of a complete “Bedside examination” considering for the analysis Nystagmus, Head Impulse Test (HIT) of the vestibulo–ocular reflex (VOR), Head Shaking test (HST), Vibratory test. We also performed the caloric bi-thermal test. All patients underwent urgent head CT scans, and only in doubtful cases, a diagnostic investigation through cranial MRI was requested.

Clinical evaluation revealed that 119 patients (51%) showed impairment in peripheral vestibular function. Among these, 58 subjects (48.7%) had a diagnosis of benign paroxysmal positional vertigo, 40 patients (33.6%) presented vestibular neuritis, 11 cases (9.2%) were described as vestibular migraine, 9 cases (7.6%) were suggestive for Ménière’s disease and only one case (0.84%) for perilymphatic fistula.

In the remaining 114 patients (49%), we could hypothesize different etiopathogenesis of acute vertigo:17 cases (14.9%) were caused by neurological diseases and 8 cases (7%) by cardiovascular impairment, in 13 patients (11.4%) a cervical origin was found, while 5 cases (4.4%) and 4 cases (3.5%) were respectively associated with iatrogenic vertigo due to drugs and psychogenic vertigo.

Symptoms resolution after one month was deduced from medical records by means of routine 30-day ENT check, while data about recovery after one year were collected by a telephone interview performed 380 days from admission to the ED. We statistically analyzed the correlation between time of resolution and comorbidities, therapies, age, sex, comorbidities, and clinical features of vertigo. Statistical analysis was performed using SSPS for Windows software. The relationship between variables was investigated by means of the Pearson Chi-Square test. The association was considered statistically significant with *p* < 0.05. 

## 3. Results

The resolution of acute vertigo attacks occurred in 189 patients (81.1%). As shown in Table 1, 135 patients (57.9%) recovered in one month both spontaneously and/or medically treated, without significant differences for sex. Among these, 91 subjects (67.4%) complained of objective vertigo, 55 (40.7%) of subjective vertigo, and 14 (10.4%) of instability. Approximately half of the early-recovered study population (45.2%) had additional audiological symptoms, and a large amount of them (57.8%) showed a neurovegetative system involvement. Clinical evaluation showed a clear prevalence of positional Nystagmus rather than spontaneous one among patients recovered at one month, with an incidence of 24.4% and 5.9%, respectively. An interesting finding is a low number (38–28.1%) of recovered patients with symptoms duration longer than 24 h. 

Moreover, we demonstrated in over 65 patients a significantly lower chance of recovery within one month than in younger ones (Table 1 and Table 2).

As shown in Table 3, the persistence of vertigo one year after ED access was reported in 44 patients (18.8%), while a recurrence was described in 52 patients (22.3%) with a different degree of severity. (*: statistically significant)

We analyzed relationship between different types of vertigo (subjective, objective, or instability), elevated blood pressure, and associated symptoms (audiological, neurological, neurovegetative symptoms, and neck pain) and recovery rate after one month (Table 1) and one year (Table 3). Statistical analysis demonstrated that the presence of instability, auditory symptoms, mostly hearing loss, neurovegetative symptoms, and neck pain represent significant factors for recovery within one year (Table 3).

We also evaluated correlations between the symptoms of patients and their clinical and pharmacological history. While a history of hypertension is associated with the worst recovery within one month, multivariate analysis showed that symptoms duration greater than one-day represents a bad prognostic factor for recovery within one month as well as one year (Table 1, Table 2 and Table 3). Moreover, symptom duration greater than one day is related to a high degree of vertigo persistence by one month and one year. Finally, age >65 years and neck pain are significantly related to symptom persistence, respectively, one month and one year after ED access (Table 4).

Interestingly, we reported significant results by vestibular assessment: patients with spontaneous Nystagmus or positive HIT during ED access showed a statistically significant difference in symptoms recovery within one month as well one year, with a higher percentage of persistence (Table 1 and Table 3).

On the contrary, the presence of positional Nystagmus is significantly involved in symptom persistence after one month, with no incidence as a bad prognostic factor for one-year recovery (Table 1).

Finally, we found vertical Nystagmus in a few patients (5 cases). This finding seems not to be significantly correlated with the prognosis at one month and one year. This Nystagmus is probably of central origin and not attributable to peripheral positional vertigo, and the lack of statistical significance could be due to the small number of cases.

## 4. Discussion

The recent literature highlights the difficulty that acute vertigo may sometimes pose to the clinician. The findings of the present study provide evidence that resolution of acute vertigo attack occurs in 81.1% of patients considered, while the remaining 18.8% show residual dizziness—as reported similarly to Heinrichs et al. [15]. In our series, recovery is age-related, and older patients recovered within one-month less than younger ones. This finding could be related to a better central compensation for young people due to their greater neurological plasticity. On the contrary, long-term recovery seems to be less related to age because of the slow central compensation that can occur in young and old patients.

In our experience, recovery by one month and one day is affected by the duration of symptoms: we hypothesize that a longer duration of vertigo indicates a greater intensity of the symptom and, therefore, a worse prognosis in the short and long term.

Regarding the recurrence of vertigo, one year after ED access, we reported a reappearance of vertigo in 22.1% of cases. We considered different vertigo as a symptom of different diseases of varying etiology; these may arise from the inner ear, brainstem, or cerebellum or may be psychiatric or internal medicine-related due to orthostatic dysregulation or toxicity for drugs and foods.

Ménière’s Disease and vestibular migraine are characterized by recurrent vertigo as defined in the Barany Society guidelines. In particular, for Ménière’s Disease, diagnostic criteria formulated by the Classification Committee of the Bárány Society in 2015 include definite Ménière’s Disease requiring the observation of an episodic vertigo syndrome associated with low- to medium-frequency sensorineural hearing loss and fluctuating aural symptoms (hearing, tinnitus and/or fullness) in the affected ear, with a duration of vertigo episodes ranging from 20 min to 12 h. On the other hand, probable Ménière’s Disease can be considered a broader concept defined by episodic vestibular symptoms (vertigo or dizziness) associated with fluctuating aural symptoms occurring from 20 min to 24 h [16]. Recently Criteria for Vestibular Migraine diagnosis has been revised, and actually, the diagnosis is based on recurrent vestibular symptoms with a duration ranging from 5 min and 72 h, history of migraine, the temporal association between vestibular symptoms and migraine symptoms, and exclusion of other causes of vestibular symptoms [17].

The recurrence rate is reported in the literature also for neuritis in 2–12% of cases [18], probably from the reactivation of a latent infection of the vestibular ganglion with herpes simplex virus type I.

In our previous papers, we describe a recurrence percentage of 30.76% for benign paroxysmal positional vertigo (BPPV) [19,20]. We highlighted the association between the recurrence of BPPV and age, female sex, and the presence of different and multiple comorbidities such as psychiatric disorders and neurological and vascular diseases [19]. We also demonstrated that various drugs are significantly associated with recurrence: antihypertensive therapy with a singular agent, central nervous system agents, PPIs, vitamin D, and thyroid hormones [20].

About clinical evolution and the particular correlation between symptom recovery and associated symptoms at the onset, we show that the possibility of healing within one year is significant for vertigo associated with auditory symptoms, mostly hearing loss. The auditory symptoms could indicate a peripheral origin of vertigo with a better long-term recovery. As described in the literature, when ipsilateral hearing loss accompanies symptoms consistent with vestibular neuritis, the site of the lesion is thought to be within the labyrinth itself, and inner-ear disorders such as labyrinthitis, labyrinthine infarction, and perilymphatic fistula should be considered [21]. In these clinical conditions, ischemia involving the common-cochlear or vestibulo-cochlear branches of the labyrinthine artery could justify vertigo with sensorineural hearing loss [22].

However, these symptoms can also be detected in disorders of the VIII cranial nerve, Ménière’s disease, herpes zoster oticus, immune or viral-mediated inner-ear disorders, or finally, in the anterior inferior cerebellar artery (AICA) stroke [23].

Similar consideration can be proposed about vestibular signs indicating peripheral vertigo. We demonstrated significant involvement in recovery from the vertigo of some clinical signs of vestibular dysfunction. In particular, the presence of Nystagmus (both spontaneous and positional) and positive HIT at the beginning are associated with a greater persistence of vertigo by one month, while the same vestibular signs indicate a better long-term recovery. It is reasonable to think that one month is not enough for central compensation, while central nervous system mechanisms could lead to an effective and complete recovery within one year.

The importance of vestibular examination has been extensively studied, and different algorithms, including both Nystagmus research and HIT, have been described for the examination of patients with acute vertigo. Two different approaches have been proposed in this setting, i.e., the HINTS test and the STANDING protocol. These methods have shown high sensitivity and specificity when performed by trained clinicians. HINTS is a bedside test for frontline clinicians to rule out stroke; it includes the Head Impulse test, Nystagmus research, and Test of Skew. Nevertheless, a recent review demonstrated that HINTS examination (if used in isolation by emergency physicians) is not sufficiently accurate to rule out a stroke in patients complaining of the acute vestibular syndrome [24]. STANDING is an acronym for the four-step examination protocol based on SponTAneous and positional Nystagmus, evaluation of the Nystagmus Direction, head Impulse test, and evaluation of the standing position and gait [9]. If performed by trained clinicians, this algorithm showed a 95% of sensitivity and 96% of specificity in diagnosing central causes of acute vertigo and seems to be more specific than the HINTS test [25]. Our clinical data are coherent with results obtained through these diagnostic protocols.

On the contrary, neurological symptoms or well-known neurologic diseases represent a negative prognostic factor for recovery, related to neurological pathogenesis of vertigo [2,12]. The literature data report that about 25% of patients with acute vertigo and dizziness have a potentially life-threatening pathology, and among these diseases, stroke occurs in 4–15% [12]. In this regard, new diagnostic index tests are proposed irrespective of the presence of Nystagmus called CATCH (central features, age, triggers, cover test with skew deviation, HIT, history of dizziness/vertigo) and EMVERT in two blocks (EMVERT block 1 including video-oculography, mobile posturography, measurement of subjective visual vertical and EMVERT block 2 with standardized MRI). Finally, among other index tests, the literature reports suggest that the neutrophil-to-lymphocyte ratio is helpful in the diagnosis of vestibular stroke and vestibular diseases [26,27].

In our experience, an additional long-term positive prognostic factor for recovery is neck pain. In the literature, cervical vertigo is still debated. However, neck disturbances combined with vertigo are commonly encountered in the clinic, and they are often associated with a good recovery [28,29].

The discussion about radiological examinations involved in the management of acute vertigo in the emergency department is one the most interesting and debated, so far still open. Personal data and results are mostly in line with the conclusion obtained by other authors. In fact, despite the disproportionately over-use of neuroimaging in patients with vertigo, this method does not correspond with the improvement in overall diagnostic yield for the central origin of vertigo. In our series, all patients were subjected to neuroimaging, but we did not find a correlation with the evolution of the disease or a certain diagnosis in most cases: overall, significant findings are present only in 5.7% of cases, including both CT (4.5%) and MRI (9.1%). Instead, in the US, about 40% of patients presenting with acute vestibular disorders undergo cranial computed tomography (CT) imaging and less than 3% MRI [30]. However, CT has a very low sensitivity (max. 16%) to identify posterior circulation stroke [31] and is mostly indicated to exclude an acute haemorrhage before intravenous thrombolysis or antiplatelet therapy. Besides, an MRI within the first 24 h after symptom onset may miss about 20% of larger strokes and up to 50% of the brainstem and cerebellar strokes with a diameter of less than 1 cm [32]. Consequently, MRI should be performed more than 48 h after symptom onset [33].

Finally, some considerations about therapy in acute vertigo syndrome are appropriate. First, it is worth noting that vertigo is not a single pathological entity but an important onset symptom of several diseases arising from the inner ear, brainstem, or cerebellum. Symptoms can also have a psychic origin, internal causes, or adverse effects of drugs. Therefore, the treatment is difficult and different specialists must be involved, including physicians of primary care to internal medicine, neurology, otorhinolaryngology, and psychiatry. Most vestibular syndromes can be treated successfully, but it is crucial to identify them. Despite the clinical importance of vertigo, patients often receive insufficient or inadequate care. Often too many drugs are prescribed, mostly ineffective and purely symptomatic. Drug treatment of acute vertigo may include central antihistamines, benzodiazepines, calcium channel blockers, antiepileptics, beta-receptor blockers, betahistine, corticosteroids, potassium channel blockers, selective serotonin reuptake inhibitors [18]. The specific therapy is closely related to the different diagnoses, and it can be conducted differently depending on the underlying pathology. In the presence of neurovegetative symptoms (nausea and vomiting), antivertigo drugs may be given to relieve symptoms in the first few days (central antihistamine, benzodiazepines); however, prolonged administration delays central compensation of the peripheral vestibular deficit (maximum period of 3 days). Recovery of peripheral vestibular function was significantly improved by monotherapy with corticosteroid [32], and BPPV can be successfully treated by means of liberatory maneuvers.

## 5. Conclusions

Although the commonest vertigo syndromes are benign, serious conditions can cause dizziness or vertigo. Our findings emphasize the need for a thorough clinical evaluation of the acutely dizzy patient, including history, comorbidities, associated symptoms, and clinical vestibular signs. They can help clinicians with central or peripheral diagnoses predicting the resolution of vertigo.

## Figures and Tables

**Table 1 jpm-13-00407-t001:** Clinical and demographic characteristics of patients and factors associated with early resolution of symptoms (1 month).

Variable	All Patients *n* = 233	Persistent at 1 Month *n* = 98	Recovered at 1 Month *n* = 135	*p* Value
Age ≥ 65 year	80 (34.3)	43 (43.9)	**37 (27.4)**	**0.009**
Sex (Male)	93 (39.9)	42 (42.9)	51 (37.8)	0.434
*ED access and symptoms*				
Classified as Urgent or Emergency	65 (27.9)	26 (26.5)	39 (28.9)	0.692
Objective vertigo	155 (66.5)	64 (65.3)	91 (67.4)	0.737
Subjective vertigo	93 (39.9)	38 (38.8)	55 (40.7)	0.762
Instability	32 (13.7)	18 (18.4)	14 (10.4)	0.080
Audiological symptoms	112 (48.1)	51 (52.0)	61 (45.2)	0.301
Hearing loss	83 (35.6)	39 (39.8)	44 (32.6)	0.257
Tinnitus	72 (30.9)	28 (28.6)	44 (32.6)	0.512
Ear fullness	40 (17.2)	14 (14.3)	26 (19.3)	0.320
Neurological symptoms	102 (44.0)	46 (46.9)	56 (41.8)	0.435
Neurovegetative symptoms	139 (59.7)	61 (62.2)	78 (57.8)	0.493
Nausea and vomiting	63 (27.0)	28 (28.6)	35 (25.9)	0.654
Neck pain	108 (46.4)	46 (46.9)	62 (45.9)	0.878
Elevated Blood Pressure	45 (19.3)	21 (21.4)	24 (17.8)	0.486
Hospital admission	19 (8.2)	10 (10.2)	9 (6.7)	0.330
*Clinical history*				
Symptoms duration > 1 day	78 (33.5)	40 (40.8)	**38 (28.1)**	**0.043**
Recent airway infection (<1 month)	11 (4.7)	2 (2.0)	9 (6.7)	0.125
Recent trauma (<1 month)	21 (9.0)	12 (12.2)	9 (6.7)	0.142
Previous episodes	11 (4.7)	3 (3.1)	8 (5.9)	0.309
History of hypertension	78 (33.5)	40 (40.8)	**38 (28.1)**	**0.043**
Cardiovascular disease (excluding hypertension)	41 (17.6)	19 (19.4)	22 (16.3)	0.541
Diabetes	14 (6.0)	7 (7.1)	7 (5.2)	0.535
Central nervous system diseases	17 (7.3)	9 (9.2)	8 (5.9)	0.345
Took medication prior to ED access	123 (52.8)	59 (60.2)	64 (47.4)	0.053
Anti-histaminic	57 (24.5)	28 (28.6)	29 (21.5)	0.214
Neuroleptic	34 (14.6)	18 (18.4)	16 (11.9)	0.164
Steroids	53 (22.7)	25 (25.5)	28 (20.7)	0.391
ASA	31 (13.3)	19 (19.4)	**12 (8.9)**	**0.020**
*Vestibular evaluation*				
Spontaneous nystagmus	25 (10.7)	17 (17.3)	**8 (5.9)**	**0.005**
Positional nystagmus	71 (30.5)	38 (38.8)	**33 (24.4)**	**0.019**
Horizontal nystagmus	42 (18.0)	26 (26.5)	**16 (11.9)**	**0.004**
Vertical nystagmus	5 (2.1)	2 (2.0)	3 (2.2)	1.000
Rotational/multidirectional nystagmus	48 (20.6)	28 (28.6)	**20 (14.8)**	**0.010**
HIT	26 (11.2)	16 (16.3)	**10 (7.4)**	**0.033**
HST	29 (12.4)	16 (16.3)	13 (9.6)	0.126
*Vibrational (96 patients)*	*16/80 (16.7)*	*7/33 (21.2)*	*9/63 (14.3)*	*0.387*
*Caloric (66 patients)*	*26/66 (39.4%)*	*15/25 (60.0%)*	*11/41 (26.8%)*	*0.001*
*Radiological findings*				
Minor abnormalities at brain imaging (any)	9 (4.3)	6 (6.7)	3 (2.5)	0.137

**Table 2 jpm-13-00407-t002:** Multivariate analysis of factors associated with the persistence of symptoms within one month).

		Wald	Sign	ODDS RATIO	95% C.I.per EXP(B)	
					Inferior	Superior
f	Age ≥ 65	8.324	*** 0.004**	2.451	1.333	4.506
	Symptoms duration ≥ 24 h	5.494	*** 0.019**	2.059	1.126	3.766
	Spontaneous nystagmus	1.095	0.295	2.365	0.472	11.854
	Positional nystagmus	0.002	0.961	1.030	0.320	3.318
	Horizontal nystagmus	1.831	0.176	2.506	0.662	9.476
	Rotational nystagmus	1.691	0.193	2.273	0.659	7.836
	HIT	0.051	0.822	0.860	0.232	3.193
	Costant	17.515	0.000	0.062		

**Table 3 jpm-13-00407-t003:** Clinical characteristics according to the resolution of symptoms at 1 year since ED access.

Variable	Persistent at 1 Year *n* = 44	Recovered at 1 Year *n* = 189	*p* Value
Age ≥ 65 years	18 (40.9)	62 (32.8)	0.308
Sex (Male)	17 (38.6)	76 (39.6)	0.848
*ED access and symptoms*			
Classified as Urgent or Emergency	10 (22.7)	55 (29.1)	0.396
Objective vertigo	23 (52.3)	113 (59.8)	0.362
Subjective vertigo	14 (31.8)	67 (35.4)	0.649
Instability	6 (13.6)	17 (9.0)	0.353
Audiological symptoms	29 (65.9)	**83 (43.9)**	**0.009**
Hearing loss	25 (56.8)	**58 (30.7)**	**0.001**
Tinnitus	14 (31.8)	58 (30.7)	0.884
Ear fullness	9 (20.5)	31 (16.4)	0.521
Neurological symptoms	24 (54.5)	78 (41.5)	0.116
Neurovegetative symptoms	35 (79.5)	**104 (55.0)**	**0.003**
Nausea and vomiting	12 (27.3)	51 (27.0)	0.969
Neck pain	29 (65.9)	**79 (41.8)**	**0.004**
Elevated Blood Pressure	9 (20.5)	36 (19.0)	0.831
Hospital admission	5 (11.4)	14 (7.4)	0.388
*Clinical history and therapy*			
Symptoms duration > 1 day	22 (50.0)	56 (29.6)	**0.010**
Recent airway infection (<1 month)	1 (2.3)	10 (5.3)	0.695
Recent trauma (<1 month)	3 (6.8)	18 (9.5)	0.773
Previous episodes	2 (4.5)	9 (4.8)	0.951
Hypertension	14 (31.8)	64 (33.9)	0.796
Cardiovascular (excluding hypertension)	8 (18.2)	33 (17.5)	0.910
Diabetes	3 (6.8)	11 (5.8)	0.802
Central nervous system diseases	6 (13.6)	11 (5.8)	0.073
Took medication prior to ED access	27 (61.4)	96 (50.8)	0.206
Anti-histaminic	14 (31.8)	43 (22.8)	0.208
Neuroleptic	6 (13.6)	28 (14.8)	0.842
Steroids	13 (29.5)	40 (21.2)	0.232
ASA	9 (20.5)	22 (11.6)	0.121
*Vestibular Evaluation*			
Spontaneous nystagmus	12 (27.3)	13 (6.9)	**<0.001**
Positional nystagmus	14 (31.8)	57 (30.2)	0.829
Horizontal nystagmus	15 (34.1)	27 (14.3)	**0.002**
Vertical nystagmus	1 (2.3)	4 (2.1)	1.000
Rotational/multidirectional nystagmus	12 (27.3)	36 (19.0)	0.224
HIT	12 (27.3)	14 (7.4)	**<0.001**
HST	9 (20.5)	20 (10.6)	0.074
*Vibrational (96 patients)*	*5/12 (41.7)*	*11/84 (13.1)*	* **0.013** *
*Caloric (66 patients)*	*9/14 (64.3)*	*16/51 (31.4)*	*0.159*
*Radiological findings*			
Minor abnormalities at brain imaging (any)	2 (5.0)	7 (4.1)	0.798

**Table 4 jpm-13-00407-t004:** Multivariate analysis of factors associated with persistence of symptoms within one year. (*: statistically significant).

	Wald	Sign	ODDS RATIO	95% C.I.per EXP (B)	
				Inferior	Superior
**Dizziness**	**0.716**	**0.157**	2.046	0.760	5.508
Audiological symptoms	0.508	0.224	1.662	0.733	3.765
Neurivegetative symptoms	0.218	0.646	1.243	0.491	3.149
Cervical pain	1.340	*** 0.004**	3.818	1.535	9.495
Symptoms duration ≥ 24 h	1.027	*** 0.008**	2.794	1.306	5.978
Spontaneous nystagmus	1.405	0.105	4.077	0.747	22.245
HIT	0.309	0.677	1.362	0.318	5.829
Horizontal nystagmus	0.414	0.505	1.513	0.448	5.113
Costant	−2.423	0.001	0.089		

## Data Availability

None.

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
