# Peer review of "Clinical Evolution of Acute Vestibular Syndrome: Longitudinal Retrospective Analysis of Epidemiological Data and Prognostic Factors for Recovery"

_jpm, 2023, doi:10.3390/jpm13030407_

Round 1
Reviewer 1 Report
As the Otolaryngologist involved in vertigo treatment, I am grateful and respect for the primary care of vertiginous patients in the Emergency Department doctors.
This article is evaluated clinical features of patients admitted to the ED with acute vertigo to establish retrospective analysis of epidemiological and prognostic factors for recovery. Outcomes of the present study would provide useful information for readers. However, data presentation is insufficient for acceptance without revision. The manuscript needs to be revised in the followings.
1. Materials and Methods
Patients were tested several equilibrium tests such as the observation of nystagmus, the (video?) Head Impulse Test, the head shaking test and the CT imaging. I think authors are possible to the differential and definitive diagnosis of peripheral vertigo with these equilibrium tests. A definitive diagnosis at the ED is required.
2. Results
The number of patients with spontaneous and positional nystagmus is the significantly difference between the vertigo persistence and the recovered (P=0.005, 0.019, respectively).
But the number of patients with vertical nystagmus is no significant difference.
Was the nystagmus for peripheral vertigo?
In patients recovered at 1 year, there are many numbers of patient with audiological symptoms or hearing loss. According to Barany criteria, did authors have patients with Meniere’s disease (MD)?
MD is characterized by recurrent vertigo attacks and patients with benign paroxysmal positional vertigo (BPPV) may experience recurrent episodes of vertigo. Have authors experienced patients with repeated visits to the ED?
3. Discussion
Line 149, 150
…., when ipsilateral hearing loss accompanies symptoms consistent with vestibular neuritis, the site of lesion is thought to be within the labyrinth itself [18,19]
In general, definitive vestibular neuritis means the acute and severe rotatory vertigo without the hearing disturbance.
What is your opinion?
As the doctor in ED, how do the authors treat patients with acute vertigo?
Please show us the roadmap for acute vertigo therapy.
What conditions or examination results lead to patient-specific treatment?
Author Response
The Authors thank the reviewer for the revision. We followed point by point your suggestion (please see the attachment).
Sincerely
PM Picciotti

Reviewer 2 Report
Congratulation for your work.
I have only one comment:
Lines 172-173: "It is known that about 25% of patients with acute vertigo and dizziness 172 have a potentially life-threatening diagnosis, including stroke in 4–15%." Kindly ask you to modify the sentence, because the meaning is ambiguous.
Author Response
The Authors thank the reviewer for the suggestion.
We modified the sentence as you suggest.
Sincerely
PM Picciotti